# Youth focused life skills training and counselling services program–An inter-sectoral initiative in India: Program development and preliminary analysis of factors affecting life skills

Gautham Melur Sukumar[1], Pradeep S. Banandur[1]*, Srividya Rudrapattana Nagaraja[2], Anusha B. Shenoy[2], Swati Shahane[2], Ravi G. Shankar[3], Arvind Anniappan Banavaram[1], Gananatha Shetty Yekkar[4], Shalini Rajneesh[5], Gururaj Gopalkrishna[6]

1 Department of Epidemiology, NIMHANS, Bengaluru, Karnataka, India, 2 Life Skills Training and Counselling Services Program, Department of Epidemiology, NIMHANS, Bengaluru, Karnataka, India, 3 Department of Health Data Science, Institute of Population Health, University of Liverpool, Liverpool, United Kingdom, 4 Government First Grade College and P G Centre, Thenkanidiyuru, Udupi, Karnataka, India, 5 Department of Youth Empowerment and Sports and Planning, Program Monitoring and Statistics Department, Government of Karnataka, Bengaluru, Karnataka, India, 6 Department of Epidemiology and Former Director, NIMHANS, Bengaluru, Karnataka, India

* docotorpradeepbs@gmail.com

## Abstract

### Objectives

This paper describes the methodology of developing and implementation of a youth focused life skills training and counselling services programme (LSTCP) and assessment of factors associated with life skills of participants pre-training.

### Design

Development of all aspects of LSTCP (modules, methods and evaluation) was through a consultative process. Experiential learning based facilitation was decided as the approach for training participants. A quasi-experimental design with pre, post and follow-up assessment post-training was finalised. Data collection was done using specifically developed semi-structured self-administered questionnaire.

### Results

Multivariable logistic regression with life skills as outcome and various exposure variables was performed. About 2/3$^{rd}$ of participants had high level of life skills (68%). Increased score of extraversion (AOR = 1.57,95% CI = 1.32–1.85), agreeableness (AOR = 1.42,95% CI = 1.16–1.73), conscientiousness (AOR = 1.9,95% CI = 1.55–2.33), physical (AOR = 1.03,95% CI = 1.01–1.04), environmental (AOR = 1.02,95% CI = 1.004–1.03) and social quality of life (AOR = 1.01,95% CI = 1.006–1.02) were associated with high life skills score.

**Data Availability Statement:** All relevant data are within the manuscript and its supporting information files

**Funding:** The author(s) received no specific funding for this work.

**Competing interests:** The authors have declared that no competing interests exist.

Higher score of neuroticism (AOR = 0.66,95% CI = 0.53–0.79) was associated with low life skills score.

## Conclusion

The results presented provide an opportunity to understand the evolution of factors affecting life skills during the follow-up of this study. This study throws light on development of LSTCP for apparently healthy population in a setting like India and its states.

## Introduction

Youth (15–29 years) comprise 27.5% of India's population [1] and contribute significantly towards sustainable continuation, socio-economic and political development of the society [2]. According to Indian national youth policy, youth have specific social, physical and mental health concerns which needs to be addressed in a timely and effective manner [1]. Nearly 87% of youth in developing countries face concerns like poverty, hunger, barriers to education, discrimination, quality health care, substance use and violence [3].

In Karnataka (a state in southern India), youth comprise 34.6% of the state's population (Census, 2011) and account for 54% of all crimes and nearly one-third of all reported suicides [4]. Alcohol consumption (17.4%) and tobacco use (35.6%) is commonly prevalent among youth [5] in the state. These preventable behavioural risks for mental disorders and Non Communicable Diseases (NCDs), often start during youth and accumulate with age. Hence, from a life-course perspective, youth are key targets for primordial and primary prevention interventions aimed to reduce injuries and NCDs, including mental disorders.

Reasons for changing behavioural risks among youth in contemporary societies are complex to delineate. Millennial youth have grown up in a different socio-cultural- environment and social support systems, as against youth in 1970s and 1980s [6]. Dwindling culturally-rooted social support mechanisms influenced by ecosystem of globalization, urbanization and economic growth in the millennium, have someway contributed to deficit of life skills necessary to lead a healthy, happy and productive life [7]. Life skills deficit, risk taking behaviour, vulnerability to behavioural risk factors leading to substance use, NCDs, mental disorders, crimes and injuries is commonly reported at multiple levels in the society, rich and poor alike. Life skills training is a proven health promotion intervention to prevent and reduce behavioural risks [8] as it imparts knowledge, facilitates attitude and skill development to support adoption of healthy decisions and behaviours [9].

The Youth Policy of Karnataka (2012) aims to empower youth by adopting a life skills approach but reaching nearly 18 million youth across the state is an operational challenge. This needed a state-wide pool of certified and quality life skills facilitators to support empowerment of youth through life skills education, through an inter-sectoral approach. To create this state-wide pool of life skills facilitators, National Service Scheme wing (NSS-a program for youth within universities in India) of the Department of Youth Empowerment and Sports, partnered with Department of Epidemiology, Centre for Public Health, [NAME OF THE INSTITUTE MASKED] to implement a training of trainers program on life skills and counselling services titled 'Developing and Implementing Youth focused Life Skills education and Counselling Services program for Youth in Karnataka'.

The program focuses on creating a pool of life skill trainers vide a 'Training of Trainers' approach. The trained officers work to facilitate a positive change in the lives of youth in

Karnataka, by empowering them with a set of essential life skills. Such a training of trainers program in such large scale lacks precedence. Thus, a methodical approach towards the design of the program for its effective implementation and evaluation was required.

This paper describes the methodology adopted to develop the overall training program along with its training modules, training schedules, assessment tool and mode of training. It also presents findings from preliminary analysis of factors associated with life skills among trainees before training (pre-training).

## Methodology

Methodology is described in two sections- A and B. Section A describes methodology of developing the life skills training program and Section B describes the methodology of testing association between participant related independent factors and their level of life skills.

### A) Developing the life skills training program and ToT module

The project team in association with subject experts conducted a series of activities [Fig 1] to develop and finalize the training components for life skills training and counselling services program (LSTCP). Training components consisted of training modules, content delivery methods, training schedules and assessment tools.

**Desk review.** An electronic literature search using key search terms ('Life skills', 'Life skills education', 'Life skills training program', 'Life skills interventions', 'Youth issues', 'Life style changes and youth') was conducted in popular databases such as PubMed, Google Scholar and EBSCO, to identify, collate and review published evidence pertaining to life skills training programs, globally and nationally. Efforts were made to access unpublished literature by contacting subject experts. Journal articles, research reports, government documents, manuals or webpages published in English language from the year 2000 onwards were included for the review.

**Stakeholder and expert workshop.** Findings from literature review which included shortlisted modules, tools and evaluation methods were discussed in a two-day stakeholder cum expert workshop. The objective of the workshop was to finalize the content, structure and mode of delivery of life skills Training including monitoring and evaluation systems. Around 15 inter-sectoral experts (from education, public health, adolescent and youth psychology, psychiatry, personality development, training, and psychiatric social work) and 15 stakeholders from NSS (volunteers, officers and coordinators), Department of Youth Empowerment and Sports, Government of Karnataka, youth and parent representatives, participated in this workshop. The shortlisted life skills modules were reviewed using participatory methods. At the end of the workshop, the UNICEF-WHO prescribed life skills were finalised to be incorporated for the study. Further, it was recommended to incorporate experiential learning as a method of delivering the workshop using a facilitatory approach. The outline of the ten modules and its contents were finalised during the workshop for the team to develop the modules.

*Module development.* A team of 20 project staff expended over 640 man-hours to develop a draft module based on suggestions of experts and stakeholders. The existing 27-day module



**Fig 1. Activities undertaken to develop life skills education and counselling services program for youth.**

used by Department of Youth Empowerment and Sports (DYES), Government of Karnataka was found most suitable for the purpose of this program. It was converted into a 6-day comprehensive life skills module, keeping in mind the time available for potential participants and resources, without significant compromise in content.

*Pilot training*. The finalised 6-day comprehensive life skills ToT module was finalized for piloting during December 2016 on 15 participants. The aim was to understand the implementation feasibility, challenges and delineate necessary logistics including budgetary requirements, manpower, and coordination mechanisms necessary for training.

*Evaluating the life skills training and counselling services program*. After a preparatory period of one year which included desk review, stakeholder cum expert workshop, module development and pilot training, the Life skills training program was started on 2nd January 2017 with 25–30 participants per batch. A quasi-experimental design was incorporated to concurrently evaluate the effectiveness of the LSTCP program. The study group involved participants who underwent life skills training for a week [NSS coordinators, officers and teaching staff]. A control group comprised of similarly qualified subjects (usually colleagues with similar working environment) who had not undergone the training. They were approached and data was collected within a week's time. A written informed consent was obtained from both study and control group subjects.

A conceptual framework of factors affecting life skills of an individual based on biopsychosocial theory proposed by George, E & Engel L (1980) was developed [Fig 2].

Other factors affecting life skills of potential participants like teaching factors, work satisfaction, quality of life etc was decided based on desk review and expert inputs during the preparatory phase. Based on this conceptual framework, a semi-structured, self-administered bilingual tool was developed with the contents based on the available literature and including different standardised scales (Table 1). This included items covering the 10 life skills domains (self-

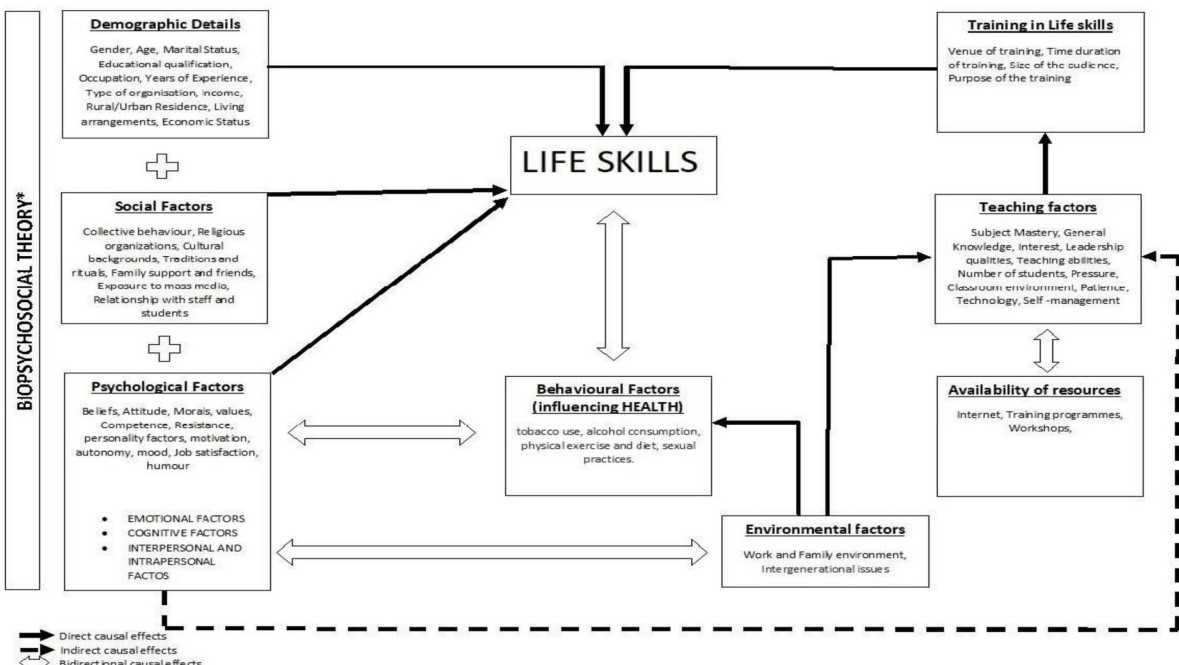

**Fig 2. Conceptual framework—factors affecting life skills and direct, indirect and bi-directional causal effects on life skills including training program.**

**Table 1. Details related to study instrument.**

| Section | Content | Operational Definition & Validation |
|---|---|---|
| Section 1: Interview Information | Auto assigned unique study number, Name, Current positions, Name of the college/institution, Contact number, Address, Taluk, District, Locale, Date, consented information. | - |
| Section 2: Editing & Data Entry | Information & details of data entry | - |
| Section 3: Sociodemographic characteristics | Sociodemographic characteristics of the respondent Household Information–Number of family members, relationship with respondent, age, occupation, education & marital status | - |
| Section 4: Family Environment | Communication with family members, Arguments, Criticism, Time spent with family members, decision making and Family support issues | - |
| Section 5: Socio economic characteristics | Own a house, Own agricultural land, Household income, Household expenditure | - |
| Section 6: Personal and Family Health | Morbidity and hospitalisation information of self and family members | - |
| Section 7: Diet and eating habits | Type of diet, consumption of number of meals cooked at home | - |
| Screening Section—8,9,10,11: Substance Use | Tobacco smoking | C.A.G.E. Questionnaire [10] |
| | Tobacco chewing | Modified Fagerstorm Tolerance scale [11] |
| | Alcohol consumption | M.I.N.I– 5.0.0 Interview [12] |
| | Injecting/sniffing/oral drugs | C.A.G.E. Questionnaire [10] |
| Section 12: Violence related information | Different forms of violence experienced, its frequency, person who inflicted violence, hospitalization due to injuries, violence inflicted by Respondent | - |
| Section 13: Depression | Symptoms of depression in last 15 days, sleep disturbance, appetite, feeling apathy, feeling worthless, lack of interest in daily activities | M.I.N.I– 5.0.0 Interview [12] |
| Section 14: Generalized Anxiety Disorder | Symptoms of anxiety–Consistently being worried, feeling restless, irritable and inability to concentrate on work. | M.I.N.I-5.0.0 Interview [13] |
| Section 15: Self-harm | Information on thoughts of committing self-harm or suicide, ever attempted to commit suicide. | M.I.N.I-5.0.0 Interview [12] |
| Section 16: Injuries and related | Different forms of injuries like road traffic accidents, falls, burns, animal bite, drowning, poisoning and hospitalization for the same. | |
| Section 17: Physical Activity | Time spent on different physical activities in a week | |
| Section 18: Sexual behaviour | Sexual practices including number of partners, use of condoms during first and last time the person had sex | |
| Section 19: Work environment and job satisfaction | Type of work, Change in jobs, Monthly income, characteristics of job satisfaction | |
| Section 20: Teaching Factors | Mode of teaching, preferred mode of teaching, perception about their teaching abilities, knowledge and technology | |
| Section 21: Peer Group & Social capital | Number of peers, activities they do with their peers and information on peer characteristics | |
| Section 22: Behavioural Factors | Self-talk, Social, family, work, financial, psychological, health related crisis, psychological, Big 5 Inventory with components of Extraversion, Openness, Conscientiousness, Agreeableness, Neuroticism. | 10-item short version of Big Five Inventory developed by Dr Rammstedt [13] |
| Section 23: Life Skills | 115 questions related to life skills | Life skills scale developed for adolescent population by Dr Vranda M N [14]. Scale validated for adult population by Dr Guatham M S [15] |
| Section 24: Quality of life | 25 questions on general health and 4 domains such as Physical quality of life, Psychological quality of life, environmental quality of life | WHOQOL–BREF Inventory developed by World health organisation [16] |
| Section 25: Exposure to media and related | Information time spent and watching programs television, using internet, video games & video Tapes | Cell phone overuse and addiction developed by Centre for Wellbeing, NIMHANS [17] |

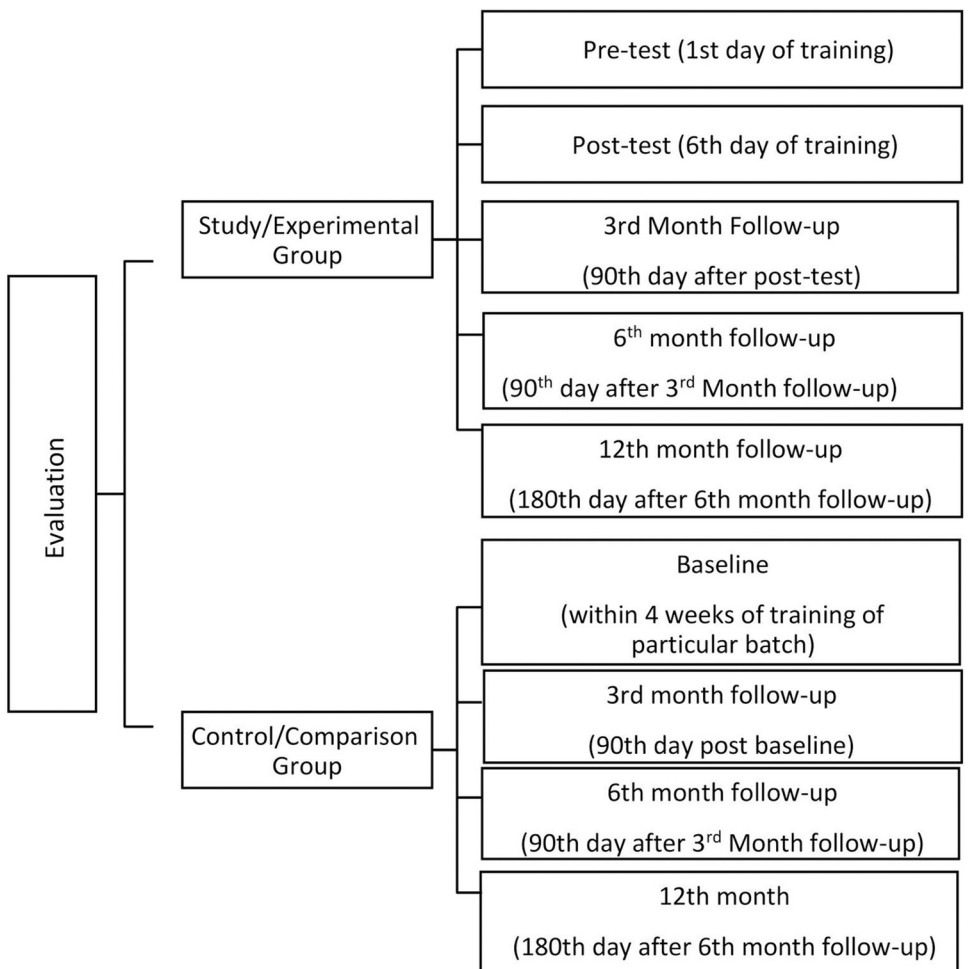

**Fig 3. Schematic representation of evaluation of the effectiveness of life skills training program.**

awareness, empathy, effective communication, interpersonal relationships, coping with stress, coping with emotions, decision making, problem solving, creative thinking and critical thinking). This tool was administered pre and post training program for study group just before and after training. Control group were administered the same tool within 4 weeks of training of study group by trained youth volunteers in their respective workplaces at a time, date and place that was mutually convenient. The follow-up assessment (post-test) was repeated for both the groups at 3rd, 6th and 12th month post training [Fig 3]. Pre-test and follow up questionnaires had 24 sections, while the post-test questionnaire had 8 sections (Factors those which assumed to not have any changes in the span of 5 days of training were excluded). Details of each section of the study tool is provided in Table 1. Life skills tool utilised for this assessment was primarily developed for adolescent population. This was validated for adult population to incorporate it for the present study. The study instrument was initially developed in English and then translated to local vernacular, Kannada by a language expert. The same was back translated to English to ensure equivalence.

Project staff facilitated the data collection process. The baseline assessment for control group as well as follow up for all participants was done at their colleges or residences as preferred by the participants in every district. Requisite number of questionnaires were sent from

the central office to the district Yuva Spandana Kendras (Youth health promotion canters) [18] on 24th of every month. Filled questionnaires from every district were sent through postal service on a weekly basis every Saturday. All data was kept confidential and stored in a locked safe with access only to designated members of the central team. Only the Principal Investigator, Project Coordinator, Research Coordinator, Statistician and Monitoring & Evaluation Officer of the study had access to these data. Although the questionnaire was self-administered, the members of the project team assisted the subjects in clarifying any questions or doubts while filling the questionnaire. The members of the project team were trained in data collection and were blinded to the training status of participants of the study. All assessments were done on both the groups during the same time. Data collection procedures were documented in a written protocol to ensure that data were collected in a standardized way. Received data was entered on a specifically designed password protected database.

## B) Testing association between participant related independent factors and their level of life skills

A total of 2037 participants are trained as on March 21 through 8 training programs. Finally, data of 1981 (97.2%) participants were utilised for analysis. Data of remaining participants were incomplete and hence excluded. The sample size was estimated based on 20% expected relative precision, 5% allowable error, expected prevalence of high life skills at 70% and expected odds ratio of 1.5 using the formula $[Z_{\alpha/2}^2 / \log^2(1-RP)] * [1/X + 1/Y]$, where, $X = 1 / \rho_p(1-\rho_p)k$, and $Y = 1 / \rho_a(1-\rho_a)$, where $Z_{\alpha/2}$ is the critical value of the Normal distribution at $\alpha/2$, RP is the relative precision, $\rho_p$ is the prevalence of the outcome in the presence group, $\rho_a$ is the prevalence of the outcome in the absence group, and k is the ratio of presences to absences being sampled $(n_p/n_a)$. Based on the calculation, minimum sample size was estimated at 814 subjects. There are 52 predictors for testing their association with level of life skills. Considering thumb rule of minimum 15 subjects for every predictor, we would require 780 subjects. Our sample of 1981 participants is sufficient. Power calculations also revealed more than 95% power.

We conducted a cross sectional analysis to assess factors associated with level of life skills among the participants of life skills training program. Data was collected from participants who attended the training. Factors tested for association with high level of life skills was included as per the developed conceptual framework [Fig 2]. Levels of life skills were the sum of scores on the likert scale of life skills questionnaire [19]. Based on the scores obtained, every participant was categorized as to have moderate (score less than 437), and high levels (score of 438 and above) of life skills.

**Statistical analysis.** Multiple logistic regression analysis was performed with level of life skills as outcome variable. Variables in the conceptual frame work were hypothesized as exposure variables. In univariate analysis, all hypothesized exposure variables associated with the outcome at 10% significance (p<0.10) were eligible to be considered for the final multiple logistic regression model. A forward stepping process was used to build the final model. Variables that were significant at 5% level (p<0.05) and those which changed the odds ratio of at least one exposure variable by 10% were eligible to be retained in the final model. The significance of addition of each exposure variable into the model was tested using likely-hood ratio test with appropriate degrees of freedom. Goodness of fit for the final model was tested using *estatgof* command followed by fitting area under the curve using *lroc* command. All the analysis was done using STATA 12.0 software for WINDOWS.

## Ethics approval

A written informed consent was obtained by the subjects of the study. All necessary ethical guidelines and principles were followed in the conduct of this study. Ethical approval for this

study was obtained from institutional ethics committee of NIMHANS vide letter No.NIM-HANS/2nd IEC (BS & NS DIV.)/2016 dated 07/12/2016

## Results

### Section A–Finalised life skills training module and training programme

We finally reviewed 77 articles to gather information regarding different modules used for youth specific life skills training and available tools to assess the effectiveness of such life skills training program. Three investigators spent nearly 500 hours (1500 man-hours) to conduct the review and shortlisted Life Skill Educators Activity Manual of 27 days' program (Level 1, Level 2 and Level 3 modules) [20] were identified and shortlisted for discussion with stakeholders and experts. The proposed youth specific life skills training ToT module includes the 10 life skills domains identified by World Health Organization [7]. It was unanimously agreed by experts and stakeholders to conduct the life skills training using adult learning principles and experiential learning through the process of facilitation based on Kolb's learning cycle [21].The approach involves four sequential steps namely, **Concrete experience** (an activity is given to participants to have an experience of the skill), **Reflective observation** (participant's opinion on the experience of the activity and learning is known), **Abstract conceptualization** (concept is delivered to the participants uniting their responses and experience of the activity), and **Active experimentation** through delivery tips (application of the experience and take home message is given to the participants on a given concept). This was adopted to enhance participant's understanding and knowledge of each life skill domain.

The final 6-day module for ToT program includes two sessions each day with one or two domain/s of Life skills imparted per session (3 hours). The last day was stipulated for demonstration and review session from participants where the participants were divided into ten groups with each group given one of the ten life skills to demonstrate as an activity-based learning session.

### Section B: Association between participant related independent factors and their level of life skills

Overall, 1981 participants attended LST during the study period. Over 2/3rd of the participants had high level of life skills (68.6%). Most participants were aged between 28–47 years (70.6%), men (72.6%), postgraduates (89.2%), married (77.6%), from urban (59.7%), practising Hinduism (92.9%), and salaried (96.7%). Age and marital status were the socio demographic variables significantly associated with level of life skills in univariate analysis [Table 2].

Most participants reported that they spend time with their family (95.9%), practice collective decision making in their families (76.2%) and worry about their concerns (42.9%). All characteristics of family environment were significantly associated with level of life skills. Among peer characteristics substance use other than alcohol and tobacco was significantly associated with level of life skills among participants in univariate analysis [Table 2].

Nearly 35.4% of participants reported to have health problems. Most participants reported to be moderate in physical activities (73.9%). Prevalence of alcohol, tobacco smoking, smokeless tobacco chewing, injecting drugs and sniffing drugs was 28.9%, 13.3%, 4.9%, 1.2% and 0.7% respectively. Risk of cell phone addiction (11.8%), feeling anxious (20.5%) and having suicidal ideation (4.7%) were found to be significantly associated with level of life skills. Participants reported that in the past six months they were exposed to violence such as being hit/kicked (1.08%), being pushed/showed (2.1%), being badly beaten up (0.25%), threatened/attacked with weapon (0.61%), being verbally/emotionally abused (10.09%), robbed (0.30%)

**Table 2. Socio-demographic characteristics, Family environment and Peer characteristics among participants attending LSTCP.**

| Characteristics | Moderate Life skills | High life skills | Total | Crude odds ratio | Confidance interval | p value |
|---|---|---|---|---|---|---|
| | n(%) | n(%) | N (%) | | | |
| **Total number of participants (N)** | 622(31.4) | 1359(68.6) | 1981(100.0) | | | |
| **Socio-demographic characteristics** | | | | | | |
| Age$^\$$(in years) | 38.7±9.1(619) | 39.8±8.6(1351) | 39.4±8.8(1970) | 1.01 | 1.00–1.03 | 0.01* |
| **Age category (in years)(N = 1970)** | | | | | | |
| 18–27 | 70(41.2) | 100(58.8) | 170 (8.6%) | | Reference | |
| 28–37 | 216(32.3) | 452(67.7) | 668 (33.9%) | 1.46 | 1.03–2.07 | 0.03* |
| 38–47 | 215(29.7) | 508(70.3) | 723 (36.7%) | 1.65 | 1.17–2.33 | 0.004* |
| 48–57 | 113(29.2) | 274(70.8) | 387 (19.6%) | 1.7 | 1.16–2.47 | 0.006* |
| > = 58 | 5(22.7) | 17(77.3) | 22 (1.11%) | 2.38 | 0.84–6.75 | 0.103 |
| **Gender(N = 1981)** | | | | | | |
| Female | 168(31.1) | 373(68.9) | 541 (27.3%) | | Reference | |
| Male | 454(31.5) | 986(68.5) | 1440 (72.6%) | 0.97 | 0.79–1.21 | 0.839 |
| **Locale(N = 1981)** | | | | | | |
| Rural | 253(31.7) | 544(68.3) | 797 (40.2%) | | Reference | |
| Urban | 369(31.2) | 815(68.8) | 1184 (59.7%) | 1.03 | 0.85–1.25 | 0.786 |
| **Religion(N = 1978)** | | | | | | |
| Others | 50(35.7) | 90(64.3) | 140 (7.07%) | | Reference | |
| Hindu | 571(31.1) | 1267(68.9) | 1838 (92.9%) | 1.23 | 0.86–1.76 | 0.25 |
| **Education(N = 1981)** | | | | | | |
| PUC | 9(36.0) | 16(64.0) | 25 (1.2%) | | Reference | |
| Degree/Diploma | 78(41.7) | 109(58.3) | 187 (9.4%) | 0.78 | 0.33–1.87 | 0.59 |
| Post-graduation & above | 535(30.2) | 1234(69.8) | 1769 (89.2%) | 1.29 | 0.57–2.95 | 0.54 |
| **Occupation(N = 1981)** | | | | | | |
| Others | 6(60.0) | 4(40.0) | 10 (0.5%) | | Reference | |
| Salaried employment | 616(31.3) | 1355(68.7) | 1971 (96.7%) | 3.29 | 0.93–11.73 | 0.065 |
| **Marital status(N = 1978)** | | | | | | |
| Never married | 163(36.9) | 279(63.1) | 442 (22.3%) | | Reference | |
| Ever married | 457(29.8) | 1079(70.2) | 1536 (77.6%) | 1.37 | 1.10–1.72 | 0.005* |
| **Family environment** | | | | | | |
| **Average Number of members in a household$^\$$** | 4±2(618) | 4±2(1350) | 4±2(1968) | 1.03 | 0.98–1.09 | 0.22 |
| **Spending time with family (N = 1978)** | 576(30.4) | 1322(69.6) | 1898 (95.9%) | 2.95 | 1.87–4.64 | <0.001* |
| **Decision making in the family(N = 1975)** | | | | | | |
| Somebody else decides | 49(50.5) | 48(49.5) | 97 (4.9%) | | Reference | |
| Participant decides | 138(37.0) | 235(63.0) | 373 (18.8%) | 1.74 | 1.11–2.73 | 0.016* |
| Collectively decide | 434(28.8) | 1071(71.2) | 1505 (76.2%) | 2.52 | 1.67–3.81 | <0.001* |
| **Worried about family members' concerns(N = 1692)** | 271(37.3) | 455(62.7) | 726 (42.9%) | 0.59 | 0.49–0.74 | <0.001* |
| **Peer characteristics** | | | | | | |
| **Substance use by peers** | | | | | | |
| Smoking tobacco(N = 1916) | 159(32.6) | 328(67.4) | 487 (25.4%) | 0.89 | 0.72–1.12 | 0.35 |
| Smokeless tobacco(N = 1910) | 84(36.1) | 149(63.9) | 233 (12.1%) | 0.77 | 0.58–1.02 | 0.072 |
| Alcohol drinking(N = 1913) | 220(32.7) | 452(67.3) | 672 (35.1%) | 0.88 | 0.72–1.08 | 0.212 |
| Substance use other than tobacco and alcohol(N = 1910) | 42(42.4) | 57(57.6) | 99 (5.1%) | 0.59 | 0.39–0.89 | 0.012 |

Row percentage; $-Mean ± standard deviation(number); n(%)—number and percentage; N- Total; Crude odds ratio—binary logistic regression

*—variable is significant at p<0.05

**Table 3. Physical health, psychological health and behavioural characteristics among participants attending LSTCP.**

| X | Moderate Life skills | High life skills | Total | Crude odds ratio | Confidence interval | p value |
|---|---|---|---|---|---|---|
| **Physical health characteristics** | | | | | | |
| | n(%) | n(%) | N (%) | | | |
| **Participants having any health problems(N = 1975)** | 228(32.5) | 473(67.5) | 701 (35.4%) | 0.91 | 0.75–1.12 | 0.401 |
| **Participants performing physical activity(N = 1958)** | 482(29.07) | 1176(70.93) | 1658 (84.6%) | 1.94 | 1.51–2.5 | <0.001* |
| **Type of physical activity(N = 1958)** | | | | | | |
| Sedentary | 55(42.0) | 76(58.0) | 131 (6.6%) | | Reference | |
| Moderate | 411(28.4) | 1036(71.6) | 1447 (73.9%) | 1.82 | 1.26–2.63 | 0.001* |
| Vigorous | 14(18.9) | 60(81.1) | 74 (3.7%) | 3.1 | 1.58–6.10 | 0.001* |
| **Average number of days participants performing physical activity in week$** | Mean±SD(n) | | | | | |
| Sedentary | 3±2(41) | 3±2(73) | 3±2(114) | 1.05 | 0.89–1.25 | 0.55 |
| Moderate | 4±3(278) | 4±3(721) | 4±3(999) | 1.05 | 0.99–1.11 | 0.08 |
| Vigorous | 3±3(11) | 2±2(42) | 2±2(53) | 0.87 | 0.65–1.17 | 0.35 |
| **Psychological health and behavioural characteristics** | | | | | | |
| **Participants** | n(%) | n(%) | N (%) | | | |
| smoking tobacco(N = 1972) | 91(34.6) | 172(65.4) | 263 (13.3%) | 0.84 | 0.64–1.108 | 0.221 |
| chewing smokeless tobacco(N = 1959) | 36(37.5) | 60(62.5) | 96 (4.9%) | 0.75 | 0.49–1.14 | 0.18 |
| drinking alcohol(N = 1955) | 187(33.1) | 378(66.9) | 565 (28.9%) | 0.89 | 0.72–1.100 | 0.29 |
| injecting drugs(N = 1964) | 9(37.5) | 15(62.5) | 24 (1.2%) | 0.76 | 0.33–1.74 | 0.51 |
| sniffing drugs(N = 1964) | 5(35.7) | 9(64.3) | 14 (0.7%) | 0.81 | 0.27–2.45 | 0.72 |
| at risk of Cell phone addiction(N = 1937) | 106(46.1) | 124(53.9) | 230 (11.8%) | 0.48 | 0.36–0.64 | <0.001* |
| feeling depressed(N = 1624) | 6(50.0) | 6(50.0) | 12 (0.7%) | 0.45 | 0.14–1.39 | 0.167 |
| feeling Anxious(N = 1760) | 150(41.4) | 212(58.6) | 362 (20.5%) | 0.59 | 0.46–0.74 | <0.001* |
| having suicidal ideation(N = 1760) | 36(42.9) | 48(57.1) | 84 (4.7%) | 0.59 | 0.38–0.92 | 0.019* |
| **Participants exposed to violence (in the past six months)** | | | | | | |
| Being Hit/kicked(N = 1942) | 10(47.6) | 11(52.4) | 21 (1.08%) | 0.49 | 0.21–1.18 | 0.115 |
| Being Pushed/showed(N = 1941) | 19(46.3) | 22(53.7) | 41 (2.1%) | 0.52 | 0.28–0.97 | 0.04* |
| Being Badly beaten up(N = 1941) | 2(40.0) | 3(60.0) | 5 (0.25%) | 0.68 | 0.11–4.11 | 0.67 |
| Being Threatened/attacked with weapon(N = 1941) | 6(50.0) | 6(50.0) | 12 (0.61%) | 0.45 | 0.15–0.41 | 0.17 |
| Being Verbally/emotionally abused(N = 1941) | 73(37.2) | 123(62.8) | 196 (10.09%) | 0.75 | 0.54–1.01 | 0.061 |
| Being Robbed(N = 1941) | 4(66.7) | 2(33.3) | 6 (0.30%) | 0.22 | 0.041–1.25 | 0.088 |
| Being Sexually harassed/Assaulted(N = 1941) | 6(40.0) | 9(60.0) | 15 (0.77%) | 0.68 | 0.24–1.93 | 0.47 |

Row percentage; $-Mean ± standard deviation(number); n(%)—number and percentage; N- Total; Crude odds ratio—binary logistic regression

*—variable is significant at p<0.05

and being sexually harassed/assaulted (0.77%). Participants who reported to be involved in physical activity, who reported being pushed/ showed were significantly associated with level of life skills in univariate analysis [Table 3].

All domains under Big 5 personality traits of participants except openness and all domains of quality of life were significantly associated with level of life skills in univariate analysis [Table 4].

In multiple logistic regression analysis, increasing personality trait scores of extraversion (AOR = 1.57, 95% CI = 1.32–1.85), agreeableness (AOR = 1.42, 95% CI = 1.16–1.73) and

**Table 4. Personality traits and quality of life among participants attending LSTCP.**

|  | Moderate Life skills | High life skills | Total | Crude odds ratio | Confidence interval | p value |
|---|---|---|---|---|---|---|
| Personality traits$ | Mean±SD(n) |  | Mean±SD(N) |  |  |  |
| Extraversion score | 3.17±0.7(602) | 3.63±0.77(1341) | 3.49±0.78(1943) | 2.19 | 1.92–2.51 | <0.001* |
| Agreeableness score | 3.64±0.6(601) | 3.94±0.67(1347) | 3.85±0.66(1948) | 2.03 | 1.74–2.36 | <0.001* |
| Conscientiousness score | 3.71±0.65(604) | 4.21±0.63(1346) | 4.06±0.67(1950) | 3.23 | 2.74–3.79 | <0.001* |
| Neuroticism score | 2.54±0.70(606) | 2.01±0.68(1346) | 2.17±0.73(1952) | 0.34 | 0.29–0.4 | <0.001* |
| Openness score | 3.13±0.44(608) | 3.12±0.42(1353) | 3.13±0.43(1961) | 0.96 | 0.77–1.21 | 0.75 |
| Quality of life$ |  |  |  |  |  |  |
| Physical quality of life | 71.38±12.69(611) | 81.81±10.79(1345) | 78.55±12.40(1956) | 1.08 | 1.07–1.09 | <0.001* |
| Psychological quality of life | 65.20±11.67(609) | 73.24±9.69(1328) | 70.71±11.01(1937) | 1.08 | 1.07–1.09 | <0.001* |
| Social quality of life | 70.53±16.73(582) | 81.95±14.05(1268) | 78.36±15.85(1850) | 1.05 | 1.04–1.06 | <0.001* |
| Environmental quality of life | 64.83±12.73(615) | 74.13±12.31(1339) | 71.20±13.17(1954) | 1.06 | 1.05–1.07 | <0.001* |

$-Mean ± standard deviation; N-Total; Crude odds ratio—binary logistic regression

*—variable is significant at p<0.05

Conscientiousness (AOR = 1.9, 95% CI = 1.55–2.33) increased the odds of having high level of life skills by 1.57 times, 1.42 times, 1.9 times respectively while increasing neuroticism was associated with decreased odds of having high level of life skills (AOR = 0.66, 95% CI = 0.53–0.79). Increasing scores of physical quality of life (AOR = 1.03, 95% CI = 1.01–1.04), social quality of life (AOR = 1.01, 95% CI = 1.006–1.02) and environmental quality of life (AOR = 1.02, 95% CI = 1.004–1.03) increased the odds of having high level of life skills by 1.03 times, 1.01 times and 1.02 times respectively. [See Table 5]. Among participants aged 28–37 years, the odds of having high life skills was 1.61 times compared to participants aged 18–27 years [Table 5].

**Table 5. Multiple logistic regression analysis of factors affecting levels of life skills among participants attending LSTCP.**

| Factors | Crude Odds ratio | 95% Confidence interval | p value | Adjusted Odds ratio | 95% Confidence interval | p value |
|---|---|---|---|---|---|---|
| Age (in years) |  |  |  |  |  |  |
| 18–27 | Reference |  |  | Reference |  |  |
| 28–37 | 1.46 | 1.03–2.07 | 0.03* | 1.61 | 1.01–2.56 | 0.045* |
| 38–47 | 1.65 | 1.17–2.33 | 0.004* | 1.54 | 0.97–2.44 | 0.63 |
| 48–57 | 1.7 | 1.16–2.47 | 0.006* | 1.27 | 0.77–2.09 | 0.35 |
| > = 58 | 2.38 | 0.84–6.75 | 0.103 | 1.94 | 0.60–6.24 | 0.27 |
| Personality traits |  |  |  |  |  |  |
| Extraversion score | 2.19 | 1.92–2.51 | <0.001* | 1.57 | 1.32–1.85 | <0.001* |
| Agreeableness score | 2.03 | 1.74–2.36 | <0.001* | 1.42 | 1.16–1.73 | 0.001* |
| Conscientiousness score | 3.23 | 2.74–3.79 | <0.001* | 1.9 | 1.55–2.33 | <0.001* |
| Neuroticism score | 0.34 | 0.29–0.4 | <0.001* | 0.665 | 0.53–0.79 | <0.001* |
| Quality of life |  |  |  |  |  |  |
| Physical quality of life | 1.08 | 1.07–1.09 | <0.001* | 1.03 | 1.01–1.04 | <0.001* |
| Social quality of life | 1.05 | 1.04–1.06 | <0.001* | 1.01 | 1.006–1.02 | 0.001* |
| Environment quality of life | 1.06 | 1.05–1.07 | <0.001* | 1.02 | 1.004–1.03 | 0.01* |

Multiple logistic regression model; *—variable is significant at p<0.05

## Discussion

LSTCP is a unique state-wide life skills training among apparently healthy participants that was developed through stakeholder and expert consultation. The life skills module, content, mode of training and evaluation tool was developed following a scientific process, inter-sectoral involvement and stakeholder consensus, before they were accepted for piloting and further use.

The cross sectional analysis of factors associated with life skills among participants of a large scale life skills training program revealed that participants with increased scores for traits of extraversion, agreeableness, conscientiousness and good physical, environmental and social quality of life was associated with high life skills. Neuroticism was associated with low levels of life skills.

The present study describes the methodology of development of LSTCP, evaluation of effectiveness as well as looking at factors associated with life skills among participants at baseline. All instruments adapted as part of data collection are known to be standardized, validated, reliable and culturally appropriate to be utilized among different population across India. Furthermore, the choice of instruments was based on desk review followed by recommendation of stakeholders and experts during the stakeholders and expert workshop followed by pilot testing of these instruments. The life skills scale utilised in this program [10] was concurrently validated for adults for assessing similar construct, as the original scale was validated only among adolescents [22].

This baseline analysis of factors associated with life skills is comprehensive in assessing different factors hypothesized as a conceptual framework. Our results of increasing scores of extraversion, agreeableness, conscientiousness, decreased neuroticism and increased quality of life increasing life skills are similar to other study findings [23]. However, the association of age reveals an interesting observation. Those participants aged 28–37 years had higher odds of increased life skills compared to 18-27-year-old participants while other higher age groups had similar odds. This needs further exploration since increasing age is known to be associated with life skills in other studies [24].

Almost 2/3rd of participants have high level of life skills pre-training. This might be due to lack of awareness among participants about their actual life skills. Participants being faculty/teachers with experience, there might be a tendency to quote their life skills as high. Further, this being a self-administered questionnaire, the data collection team may not have sufficient control over the responses provided by participants. Briefing the participants about the purpose of collecting data, informed consent process and availability of project staff to clarify questions might have minimised such a possibility. However, this cannot be ruled out. As in any other health promotion program, it is likely that people with high level of life skills attend these programs compared to those with lower life skills (healthy worker effect), more so with the mechanism of deputation of participants for training. The length of the questionnaire is likely to induce fatigue while completing the questionnaire [25]. Due care was taken to organise the questionnaire to follow logical order and flow along with its layout and format. Respondent fatigue was specifically looked into during the pilot phase and addressed. Further, our experience conducting data collection doesn't indicate that the respondents experienced fatigue.

The programme LSTCP is the first such large scale inter-sectoral driven life skill and positive mental health development programme on apparently healthy individuals in India. To our knowledge, there is no such programs in terms of scale and population elsewhere. Stakeholder and expert consultation in need-based programme development is a well-established and accepted model to ensure a sustainable and focussed programme [26]. This was done due to

lack of precedence of life skills programs implemented elsewhere among apparently healthy population. Most life skills programs are focused on persons living with HIV [27], vulnerable and marginalized groups like migrants [28]. This life skills training program was required to be sustainable, implementable among participants who are apparently healthy and participants from different geo-cultural experiences from across 30 districts of Karnataka.

Adult experiential learning methods and facilitatory approach were incorporated into LSTCP programme due to following reasons- (i) target trainees were teachers; (ii) life skills does not have a simple 'Right' or 'Wrong' answer for a situation; (iii) it requires learning from group opinions, experiences and ideas; In addition, the experts recommended to focus on the affective (attitude) domain of learning. Facilitation as an approach, is known to enable participants to share ideas, opinions, experiences, and expertise in order to achieve a common goal [29] and serve as resource for learning, ensuring the sessions to be unique as it evolves through sharing life experiences, renders the environment active and conducive for participatory experiential learning environment. This helps improve the attitudinal component of participants keeping a well-defined broader structure of life skills training and methodologies intact. Further, experiential learning techniques favours focusing on affective and psychomotor domains, and 'learning by doing' were included as content delivery methods [30].

Most known life skills programs are focused on persons living with HIV [27], vulnerable and marginalized groups like migrants [28]. This program is unique and focussed on training of trainers to cater to a large youth population across the state. With 1981 participants trained in life skills, this inter-sectoral collaborative model has ensured availability of at least 50 trained manpower available to serve the cause of youth mental health promotion within each of the 30 districts of Karnataka. This program exemplifies inter-sectoral approaches envisaged in the primary health care model to deliver mental health promotion services for youth in the state.

The evaluation of effectiveness of this training program on improving life skills and wellbeing is one of the strengths of the program. Generally, programme evaluation of life skills training is limited to Pre and Post-test evaluation. The evaluation in LSTCP has a follow-up of one year. This is a major strength of the program. Follow-up evaluation provides an opportunity to understand the refresher or re-training needs of participants. Untrained colleagues as participants for control group seems to be the most appropriate choice given the feasibility and practicability of evaluation. Neighbourhood, sibling or relatives as controls was also considered. However, stakeholders and experts recommended colleagues could be more appropriate controls both epidemiologically and operationally. Evaluation of effectiveness of life skills training would have been ideal with a randomised controlled trial design. However, with the participants being deputed from different universities and directorates, randomisation was not possible. Thus, a quasi-experimental design was adopted as a feasible study design. The project team could have enrolled the participants directly but, official deputation provides administrative commitment as well as ensures participation for the training program. The possibility of healthy worker effect is likely initially into the training programmes. However, with long duration of the program implementation and large sample size, this seems unlikely.

Life skills in an individual is affected by different social, biological and psychosocial factors (Fig 2) that in turn affect health through complex interactions according to Bio-psychosocial theory [31]. In addition, life skills are known to be associated with behavioural and teaching factors, work environment and self-efficacy. Teaching factors are also influenced by the availability of resources and vice versa. The ecological perspective focuses on the individual and the interrelationship with his or her environment. This interrelationship exists between the individual and others in particular geographic and socially-constructed environments (or systems), including the individual, group, family, community, institutions, class, and culture [32]. In addition, contextual factors like venue of the training will have a direct impact on Life skills

development [33]. Thus, all these factors were considered to be confounding the effect of training on life skills and included as part of the tool for data collection.

Development and implementation of any training program is not without limitations and challenges. One important challenge would be that participants of control group may get trained eventually over time. Enquiry into such possibility is incorporated in the instrument used for follow-up data collection. This makes it possible to adjust for such change during the analysis. Furthermore, it serves as an opportunity to assess these individuals as their own controls. Contamination due to sharing of training experience among participants in control group through trained participants is another limitation that cannot be ruled out. This is likely to enrich training experience of the participants, since, facilitation and experiential learning process is based on sharing participant ideas, opinions, experiences and expertise and provides another perspective for the overall learning experience of the participants. The present study employs a quantitative approach to look at factors associated with life skills. Further studies are recommended to look at changes from a qualitative perspective which helps individuals to subjectively improve life skills to deal effectively with their everyday life demands and challenges.

## Conclusion

Despite its limitation and challenges, the LSTCP is a large scale state wide health promotion program focused on youth through their NSS officers, coordinators, and teachers within the colleges and universities in Karnataka. This program has the potential to reach at least 70% of youth in Karnataka, who are in colleges and universities. This will pave the way to equip youth in Karnataka with the essential 21st century skills that is considered one of the grand challenges of current day youth. The present study throws light on how life skills training program was developed for apparently healthy population in a setting like India with implications for other states in country in specific and any other country with a similar socio-cultural environment in general. It will also be interesting to see the way personality traits and quality of life that are found to be affecting life skills pre-training, evolves during the period of follow-up of this study. Further analysis will elucidate effectiveness of training program in immediate, short and long term over a period of one year providing insights into training and re-training needs as well as its effectiveness in response to real life situations.

## Supporting information

**S1 Checklist. TREND statement checklist.**
(PDF)

**S1 File. Protocol for data collection for the program life skills training and counselling services program.**
(DOCX)

## Acknowledgments

Authors would like to express their gratitude to, State NSS wing, department of Youth Empowerment and Sports, Government of Karnataka along with the experts and stakeholders who participated in the development of LSTCP. We would like to extend our gratitude for the study participants for taking part in the program.

## Author Contributions

**Conceptualization:** Gautham Melur Sukumar, Pradeep S. Banandur, Srividya Rudrapattana Nagaraja, Swati Shahane, Gururaj Gopalkrishna.

**Data curation:** Srividya Rudrapattana Nagaraja, Anusha B. Shenoy.

**Formal analysis:** Anusha B. Shenoy, Ravi G. Shankar.

**Funding acquisition:** Pradeep S. Banandur, Gananatha Shetty Yekkar, Shalini Rajneesh.

**Investigation:** Srividya Rudrapattana Nagaraja.

**Methodology:** Gautham Melur Sukumar, Pradeep S. Banandur, Srividya Rudrapattana Nagaraja, Anusha B. Shenoy, Swati Shahane.

**Project administration:** Gautham Melur Sukumar, Pradeep S. Banandur, Swati Shahane, Arvind Anniappan Banavaram.

**Software:** Srividya Rudrapattana Nagaraja, Anusha B. Shenoy.

**Supervision:** Gautham Melur Sukumar, Pradeep S. Banandur, Swati Shahane, Arvind Anniappan Banavaram.

**Validation:** Anusha B. Shenoy.

**Writing – original draft:** Srividya Rudrapattana Nagaraja.

**Writing – review & editing:** Gautham Melur Sukumar, Pradeep S. Banandur, Srividya Rudrapattana Nagaraja, Swati Shahane, Arvind Anniappan Banavaram.

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
