## [Decision Letter · Decision Letter 0]

2 Feb 2023

PONE-D-22-18646Youth focused Life Skills Training and Counselling Services program–an inter-sectoral initiative in India: Program Development and Preliminary analysis of factors affecting life skillsPLOS ONE

Dear Dr. Gautham M S, 

Thank you for submitting your manuscript to PLOS ONE. After careful consideration, we feel that it has merit but does not fully meet PLOS ONE’s publication criteria as it currently stands. Therefore, we invite you to submit a revised version of the manuscript that addresses the points raised during the review process. You are requested to go through the comments by the reviewer and editor and address the same. In order to improve readership, concise the article. 

We look forward to receiving your revised manuscript.

Kind regards,

Muralidhar M. Kulkarni

Academic Editor

PLOS ONE

Journal Requirements:

b) If there are no restrictions, please upload the minimal anonymized data set necessary to replicate your study findings as either Supporting Information files or to a stable, public repository and provide us with the relevant URLs, DOIs, or accession numbers. For a list of acceptable repositories, please see http://journals.plos.org/plosone/s/data-availability#loc-recommended-repositories

Additional Editor Comments (if provided):

Dear Dr. Gautham,

The article is well written, however it needs to be refined as per the suggestions below.

Reviewers' comments:

Reviewer's Responses to Questions

**Comments to the Author**

1. Is the manuscript technically sound, and do the data support the conclusions?

Reviewer #1: Yes

2. Has the statistical analysis been performed appropriately and rigorously? 

Reviewer #1: Yes

3. Have the authors made all data underlying the findings in their manuscript fully available?

Reviewer #1: Yes

4. Is the manuscript presented in an intelligible fashion and written in standard English?

Reviewer #1: Yes

5. Review Comments to the Author

Reviewer #1: The authors have appropriately addressed an important research question in the manuscript.

Following queries need to addressed.

Lines 76-77: Please rephrase the statements.

Lines 93-100: Have not been referenced

Lines 146-155: More details could be provided on conduct & outcomes of the workshop

Line 169: What was done during the preparatory period of one year?

Lines 175-177: The statements need to be rephrased. More details need to be provided about the selection & timing of control group.

Line 185: Bilingual tool – development, translation & validation details to be mentioned

Line 255-258: Written informed consent needs to be included

Table2: What does -worried about their concerns mean?

Table 3: were the participants able to clearly define – feeling depressed, feeling anxious & having suicidal ideation?

Table 4: Details about different traits & quality of life components to be provided in methods or provided as supplementary material

6. PLOS authors have the option to publish the peer review history of their article (what does this mean?). If published, this will include your full peer review and any attached files.

Reviewer #1: No

---

## [Author Response · Author response to Decision Letter 0]

8 Feb 2023

Journal Requirements:

Response: Thank you, the template links were helpful. We have re-checked the PLOS ONE’s style requirements, including those for file naming according to the journal templates and made necessary changes.

Response: We have added details about written informed consent obtained by participants in the methods section (Line: 186) as well as mentioned it under ethics statement (Line: 270) in the manuscript as well as made the changes in online submission information. Thank you. 

Response: Details provided in the cover letter. 

b) If there are no restrictions, please upload the minimal anonymized data set necessary to replicate your study findings as either Supporting Information files or to a stable, public repository and provide us with the relevant URLs, DOIs, or accession numbers. For a list of acceptable repositories, please see http://journals.plos.org/plosone/s/data-availability#loc-recommended-repositories

Response: Please see response to 3a above.

Response: Thank you we have reviewed the reference list and made necessary changes

Reviewers' comments:

Reviewer's Responses to Questions

Comments to the Author

1. Is the manuscript technically sound, and do the data support the conclusions?

Reviewer #1: Yes

2. Has the statistical analysis been performed appropriately and rigorously?

Reviewer #1: Yes

3. Have the authors made all data underlying the findings in their manuscript fully available?

Reviewer #1: Yes

4. Is the manuscript presented in an intelligible fashion and written in standard English?

Reviewer #1: Yes

5. Review Comments to the Author

Reviewer #1: The authors have appropriately addressed an important research question in the manuscript.

Lines 76-77: Please rephrase the statements.

Response: Thank you, this has now been addressed. (Line: 77)

Lines 93-100: Have not been referenced

Response: Thank you, the reference for the same has been added.

Lines 146-155: More details could be provided on conduct & outcomes of the workshop

Response: Lines: 157 to 160 - Details on the conduct & outcomes of the workshop is now provided. This has enriched the content of the manuscript (Line: 157 to 160) . Thank you.

Line 169: What was done during the preparatory period of one year?

Response: During the preparatory period of one-year desk review, stakeholder cum expert workshop, module development, pilot training and evaluation methods were finalised. The same is also included in the article. (Lines: 174 to 175)

Lines 175-177: The statements need to be rephrased. More details need to be provided about the selection & timing of control group.

Response: Thank you the mentioned statements are rephrased and added details about the selection and timing of control group. (Lines: 179 – 182)

Line 185: Bilingual tool – development, translation & validation details to be mentioned

Response: The contents of the tool was based on the available literature and including different standardised scales. The same is also included in the manuscript. (Lines: 195-196 and 209-211)

Line 255-258: Written informed consent needs to be included

Response: This is included in the methods (Lines: 186 & 270). Thank you. 

Table2: What does -worried about their concerns mean?

Response: Thank you for this important observation. We have now changed this to “Worried about Family members’ concerns” 

Table 3: were the participants able to clearly define – feeling depressed, feeling anxious & having suicidal ideation?

Response: Yes, as the statements in the tool clearly defines it as well as the statements leading to it defines it. Also, as there were trained project staff during the tool administration the clarifications were provided for the participants while responding to those statements. 

Table 4: Details about different traits & quality of life components to be provided in methods or provided as supplementary material

Response: This is provided in table 1 of the manuscript and also will be provided as supplementary material

6. PLOS authors have the option to publish the peer review history of their article (what does this mean?). If published, this will include your full peer review and any attached files.

Do you want your identity to be public for this peer review? For information about this choice, including consent withdrawal, please see our Privacy Policy.

Reviewer #1: No

Editors comments

Introduction:

1. Line 87: Authors mention tobacco use (35.6%), however the reference no. 5 does not have a link and the citation will not lead to appropriate reference. Pls verify.

Response: Thank you for noticing. The reference is now complete. 

Methodology

1. Page 7. Line 143 to 145: We finally reviewed 77 articles to gather information regarding different modules used for youth specific Life skills training and available tools to assess the effectiveness of such life skills training program.

Needs to be mentioned in results section

Response: This is included in the results sections (Lines: 276 to 278). Thank you. 

2. Table 1: Section 1, Interview information. The points are not clear, need to elaborate. 

Response: The points under section 1 interview information is elaborated in Table 1

3. Pg 14. Line 216. The Data collection procedures documented in a written protocol can be provided as supplementary material for reference of readers. 

Response: The written protocol on data collection procedures is provided as supplementary file

4. Pg. 15, Line 222 -223. Rephrasing required. 

Response: Thank you. These sentences are rephrased in manuscript (Lines: 234 to 240)

Discussion:

1. Repetition can be avoided. Line 377 to 379 has been mentioned in earlier section also. 

Response: Thank you for the observation. This is avoided in the discussion section. (Lines: 391to 393)

2. Pg 29. Line 440 to 449. It would be better to restrict the findings to present study. 

Response: The above mentioned points relate to the strengths and limitations of the methodology adopted (and not a finding of this study) for the present study like control group getting trained eventually, contamination etc. (Lines: 454 – 4643). We feel that this is important to be retained. We request you to kindly consider retaining the same. Thank you.

References

1. Ensure links are working.

Response: Thank you for noticing this. The links are re-checked and the provided links are working

2. Ref 5, 7 and 22 needs to be as per the standard guidelines. 

Response: The mentioned references are re-checked and made necessary changes as per the standard guidelines.

---

## [Decision Letter · Decision Letter 1]

10 Apr 2023

Youth focused Life Skills Training and Counselling Services program–an inter-sectoral initiative in India: Program Development and Preliminary analysis of factors affecting life skills

PONE-D-22-18646R1

Dear Dr. Banandur S Pradeep,

We’re pleased to inform you that your manuscript has been judged scientifically suitable for publication and will be formally accepted for publication once it meets all outstanding technical requirements.

Kind regards,

Muralidhar M. Kulkarni

Academic Editor

PLOS ONE

Additional Editor Comments (optional):

The article is accepted for publication.

Reviewers' comments:

Reviewer's Responses to Questions

**Comments to the Author**

1. If the authors have adequately addressed your comments raised in a previous round of review and you feel that this manuscript is now acceptable for publication, you may indicate that here to bypass the “Comments to the Author” section, enter your conflict of interest statement in the “Confidential to Editor” section, and submit your "Accept" recommendation.

Reviewer #1: All comments have been addressed

2. Is the manuscript technically sound, and do the data support the conclusions?

Reviewer #1: Yes

3. Has the statistical analysis been performed appropriately and rigorously? 

Reviewer #1: Yes

4. Have the authors made all data underlying the findings in their manuscript fully available?

Reviewer #1: Yes

5. Is the manuscript presented in an intelligible fashion and written in standard English?

Reviewer #1: Yes

6. Review Comments to the Author

Reviewer #1: The authors have satisfactorily addressed all the queries of the reviewers. The article is now acceptable.

7. PLOS authors have the option to publish the peer review history of their article (what does this mean?). If published, this will include your full peer review and any attached files.

Reviewer #1: No

---

## [Editor Report · Acceptance letter]

10 May 2023

PONE-D-22-18646R1 

Youth focused Life Skills Training and Counselling Services program–an inter-sectoral initiative in India: Program Development and Preliminary analysis of factors affecting life skills 

Dear Dr. Banandur:

I'm pleased to inform you that your manuscript has been deemed suitable for publication in PLOS ONE. Congratulations! Your manuscript is now with our production department. 

Kind regards, 

on behalf of

Dr. Muralidhar M. Kulkarni 

Academic Editor

PLOS ONE